# The Genetic Stability, Replication Kinetics and Cytopathogenicity of Recombinant Avian Coronaviruses with a T16A or an A26F Mutation within the E Protein Is Cell-Type Dependent

**DOI:** 10.3390/v14081784

**Published:** 2022-08-15

**Authors:** Isobel Webb, Sarah Keep, Kieran Littolff, Jamie Stuart, Graham Freimanis, Paul Britton, Andrew D. Davidson, Helena J. Maier, Erica Bickerton

**Affiliations:** 1The Pirbright Institute, Woking GU24 0NF, UK; 2School of Cellular and Molecular Medicine, Faculty of Life Sciences, The University of Bristol, Bristol BS8 1TH, UK

**Keywords:** avian coronavirus, cell type, envelope protein, chicken, viroporin

## Abstract

The envelope (E) protein of the avian coronavirus infectious bronchitis virus (IBV) is a small-membrane protein present in two forms during infection: a monomer and a pentameric ion channel. Each form has an independent role during replication; the monomer disrupts the secretory pathway, and the pentamer facilitates virion production. The presence of a T16A or A26F mutation within E exclusively generates the pentameric or monomeric form, respectively. We generated two recombinant IBVs (rIBVs) based on the apathogenic molecular clone Beau-R, containing either a T16A or A26F mutation, denoted as BeauR-T16A and BeauR-A26F. The replication and genetic stability of the rIBVs were assessed in several different cell types, including primary and continuous cells, ex vivo tracheal organ cultures (TOCs) and in ovo. Different replication profiles were observed between cell cultures of different origins. BeauR-A26F replicated to a lower level than Beau-R in Vero cells and in ovo but not in DF1, primary chicken kidney (CK) cells or TOCs. Genetic stability and cytopathic effects were found to differ depending on the cell system. The effect of the T16A and A26F mutations appear to be cell-type dependent, which, therefore, highlights the importance of cell type in the investigation of the IBV E protein.

## 1. Introduction

Coronaviruses (CoVs) are a highly diverse family of enveloped viruses with large positive-sense single-stranded RNA genomes that belong to the order *Nidovirales,* suborder *Coronavirinae*, family *Coronaviridae,* subfamily *Orthocoronavirinae*. Within the subfamily, there are four genera, *Alphacoronavirus*, *Betacoronavirus*, *Gammacoronavirus* and *Deltacoronavirus.* In humans, the *Alphacoronavirus* human coronavirus (HCoV) 229E and related coronaviruses are responsible for 18% of common cold cases [1]; however, infection with the *Betacoronaviruses* severe acute respiratory syndrome CoV (SARS-CoV), severe acute respiratory syndrome CoV-2 (SARS-CoV-2) and Middle East respiratory syndrome coronavirus (MERS-CoV) can lead to more serious respiratory disease that can result in mortality [2,3,4]. In animals, the *Deltacoronavirus* porcine deltacoronavirus (PDCoV), which was first identified in 2012 [5], causes enteric disease within pigs [6]. Infectious bronchitis virus (IBV) is the prototype *Gammacoronavirus* and causes the highest direct disease costs to the poultry industry within the UK and across the globe [7]. IBV infects the epithelial surfaces within the trachea, oviduct and kidney of poultry and can cause clinical signs ranging from mild respiratory signs, including snicking and tracheal rales, to severe kidney and oviduct disease [8]. As a result, it reduces egg production and quality in layers, causes reduced weight gain in broilers and leaves chickens vulnerable to secondary bacterial infections which may be fatal [9].

The IBV genome, similarly to all other CoVs, encodes four structural proteins: Spike (S), Nucleocapsid (N), Membrane (M) and Envelope (E) [10]. The CoV E protein is a small ~12 kDa protein, present in low quantities in the viral envelope [11]. The IBV E protein is present in two distinct forms during infection: monomeric and pentameric [12]. These two separate forms are thought to have different roles during infection. The monomer interacts with cellular proteins to alter the secretory pathway, and the pentamer forms an ion channel (IC) in planar lipid bilayers with low or poor selectivity [13]; viral proteins, which act as ICs, are described as viroporins. Prior studies have proposed roles for CoV E proteins in assembly [12,14,15], viral release [16,17,18] and pathogenesis [19,20,21]. CoVs assemble and bud intracellularly at the ER-Golgi intermediate compartment (ERGIC), and it has been reported during IBV infection that the majority of E remains localised in the membranes of the Golgi complexes [22]. Research suggests that both M and E proteins of IBV are required for viral budding and the formation of the viral envelope, as both E and M are reported to be necessary for the formation of virus-like particles (VLPs) [23]. Additionally, the E protein is thought to facilitate scission, as mutations introduced into the E protein of the *Betacoronavirus* mouse hepatitis virus (MHV) resulted in elongated virions [14]. The role of the E protein in pathogenesis is thought to be facilitated by the C-terminal domain, which has been shown in SARS-CoV to cause the overexpression of inflammatory cytokines [21].

The first discovered viroporin was found within the encephalomyocarditis (EMC) virus, as it was able to induce membrane permeability within infected cells [24], an activity identified for the SARS-CoV E protein [25]. Mutations generated either at residue N15 or V25 within the transmembrane domain of the SARS-CoV E protein abolish IC activity [26]. Mutation of equivalent residues in IBV, T16 and A26 reportedly caused the same effect [16], and the effect of these mutations has been predominantly characterised within continuous cell lines that are not relevant to IBV natural infection. The V25 residue of the SARS-CoV E protein is located between the monomer–monomer interface of the E homo-pentamer [20]. Mutation of the corresponding A26 residue in the IBV E protein prevents homodimerisation [12], this residue is required for the formation of VLPs [12]. The T16 residue forms the monomeric IBV E protein [12], a T16A mutation at this residue has been shown to prevent the disassembly of the Golgi [12,27] and cause premature cleavage of the S protein [28]. A separate study, however, proposed that it is the T16 residue and not A26 which is essential for homodimerisation [16]. Regardless, recombinant SARS-CoV containing the N15A mutation resulted in reduced clinical signs in mice, indicating that the E protein is a virulence factor and a target for vaccine development [20].

In this study, we characterised the effect of the two mutations, T16A and A26F, on the replication of IBV in cell systems relevant to IBV natural infection, including primary chicken kidney (CK) cells and ex vivo tracheal organ cultures, with the aim of determining whether the T16 and A26 residues were vital for IBV replication. Using a vaccinia virus-based reverse genetics system [29], we generated recombinant IBVs (rIBVs) based on the apathogenic molecular clone, Beau-R, that contained either the mutation T16A or A26F within the E protein. The replication of the rIBVs was assessed in continuous cell lines Vero and DF1, primary CK cells, and in ovo and in ex vivo tracheal organ cultures (TOCs). Different replication profiles were observed between cell cultures of different origins. The genetic stability of the mutations was also assessed in CK cells and in ovo and was also found to differ depending on the cell system. This report furthers previous research and the current understanding of E protein in IBV replication through the characterisation of the rIBVs in relevant cell systems to IBV natural infection and highlights the cell-dependent effect of the T16A and A26F mutations.

## 2. Materials and Methods

### 2.1. Cells and Viruses

All cells were provided by the Central Services Unit at the Pirbright Institute. The cell cultures were maintained within incubators set at 37 °C with 5% carbon dioxide (CO_2_). Primary CK cells were extracted from 2–3 week old specific pathogen-free (SPF) Rhode Island Red (RIR) chickens following a previously described protocol [30]. DF1 cells are a continuous avian cell line of chicken embryo fibroblasts derived from 10-day-old East Lansing eggs [31]. Vero cells are a continuous cell line derived from kidney epithelial cells of the African green monkey. Embryonated VALO hens’ eggs were sourced from VALO BioMedia GmbH. TOCs were generated from trachea extracted from 19-day-old SPF embryos using a method previously described [32]. 

The rIBV Beau-R has been described previously [29]. Beau-R is a molecular clone of the non-pathogenic IBV strain Beau-CK (GenBank Accession number AJ311317). Beau-R was propagated in 10-day-old SPF embryonated hens’ eggs, and allantoic fluid was harvested 24 hours (h) post infection (hpi) and clarified using low-speed centrifugation. All genome positions are in relation to GenBank accession number AJ311317. 

### 2.2. Construction of rIBVs Containing the T16A and A26F Mutations

Two individual point mutations were generated within rIBV Beau-R using an established vaccinia-virus-based reverse genetics system [29,33]. The T16A mutation was created via a single nucleotide mutation A24246G and A26F was generated through three-point mutations from 24276 to 24278, changing the nucleotide sequence from GCA to TTT. These mutations are identical to those generated within a Beaudette backbone, which were described as IC inactive, but the conductivity data was not shown [16]. Three isolates, denoted as 1, 2 or 3, of each rIBV were successfully rescued and were passaged in primary CK cells. Viral stocks were propagated within primary CK cells with the supernatant harvested at 24 hpi.

### 2.3. Full Genome Sequencing of Viral Stocks

BeauR-T16A and BeauR-A26F, 4 mL of each isolate, were concentrated using a 30% sucrose cushion and ultracentrifugation for 1 h at 223,600× *g*—as described [34]. RNA was extracted from the resulting pellet using a Qiagen RNeasy mini kit following the manufacturer’s protocol. The RNA concentration was quantified using the Qubit fluorometer RNA Assay HS kit. Next Generation Sequencing (NGS) was carried out by the Sequencing Unit at The Pirbright Institute, following a method detailed previously [35]. 

### 2.4. Titration of IBV via Plaque Assay and Quantification of Plaque Size

Plaque assays were carried out using CK cells as previously described [36]. All samples were titrated in triplicate. Plaque sizes were measured from ten plaques per biological repeat using NIH ImageJ software [37], totalling 30 plaques per virus. 

### 2.5. Assessment of Viral Release

CK or Vero cells were seeded into 6-well plates to confluency. Cells were infected with 500 µL of Beau-R, BeauR-T16A or BeauR-A26F isolates at a titre of 1 × 10^4^ plaque forming units (PFU) and incubated for 1 h at 37 °C with 5% CO_2_. After incubation, the inoculum was removed, and the cells were washed twice with PBS, followed by the addition of 3 mL of N, N-bis [2-hydroxethy1]-2-aminoethanesulfonic acid (BES) media [33]. At 24 hpi, the supernatant was removed from the cells and stored at −80 °C. The plates were washed once with PBS, and 1 mL of BES media was added to the cells before the plates were freeze-thawed to release the virus (cell lysate). The quantity of infectious progeny in both the harvested supernatant and cell lysate was determined in triplicate via plaque assay in CK cells.

### 2.6. Viral Replication Kinetics Assessment In Vitro

Confluent CK, DF1 or Vero cells, seeded in 6-well plates were infected with 500 µL of Beau-R, BeauR-T16A or BeauR-A26F isolates at a titre of either 1 × 10^4^ or 1 × 10^5^ PFU and incubated at 37 °C for 1 h with 5% CO_2_. The virus was removed, and the cells were washed twice with PBS followed by the addition of 3 mL BES media. The supernatant containing the infectious progeny was harvested at either 1, 24, 48, 72 and 96 hpi, to establish replication over multiple rounds of viral replication, or at 1, 2, 4, 6, 8, 10 and 11 hpi, to assess replication over the first round of viral replication. The quantity of infectious progeny was quantified in triplicate via plaque assay in CK cells.

### 2.7. Ciliary Activity Assessment in Ex Vivo TOCs

TOCs, singly plated in glass tubes, were washed twice with PBS. In replicates of ten, each TOC was inoculated with 500 µL TOC infection medium [30], containing 1 × 10^4^ PFU of Beau-R, BeauR-T16A or BeauR-A26F isolates. Mock infected TOCs were inoculated with 500 µL TOC medium. Tubes were incubated upright for 1 h at 37 °C, after which the inoculum was removed and the TOCs washed twice with PBS. 1ml of TOC infection media was added per tube, and the TOCs were incubated at 37 °C, 1 revolution per 7 min. The ciliary activity of each TOC was assessed at 24 h intervals using a light microscope, and the percentage of cilia beating was calculated as described [38,39].

### 2.8. Viral Replication Kinetics Assessment in Ex Vivo TOCs

Six TOCs were plated per glass tube and were washed twice in PBS. Each tube was inoculated with 500 µL TOC infection medium containing 1 × 10^4^ PFU of Beau-R, BeauR-T16A-1, or BeauR-A26F-3. Tubes were incubated upright for 1 h at 37 °C after which the inoculum was removed and the TOCs washed twice with PBS. 1 mL of TOC infection media was added per tube, and the TOCs were incubated at 37 °C, 1 revolution per 7 min. The supernatant from each tube was harvested at 1, 24, 48, 72 and 96 hpi, and the quantity of infectious progeny was quantified in triplicate via plaque assay in CK cells.

### 2.9. Virus Replication Kinetics In Ovo

A protocol for propagation of IBV in eggs has been described [40]. In replicates of three, 10-day-old SPF embryonated hens’ eggs were inoculated via the allantoic cavity with either 1 × 10^4^ or 1 × 10^5^ PFU of either Beau-R, BeauR-T16A or BeauR-A26F isolates. After 24 h, embryos were culled by refrigeration for a minimum of 4 h, and the allantoic fluid from each egg was harvested. The allantoic fluid was clarified using low-speed centrifugation. The quantity of infectious progeny was determined via plaque assay in triplicate in CK cells. The sequence of the E gene was determined as described in Section 2.11.

### 2.10. Serial Passaging of rIBVs in CK Cells

Confluent CK cells seeded into 6-well plates were washed once with PBS. In replicates of four, 500 μL of BES medium containing neat rIBV was added per well and incubated for 1 h at 37 °C. The inoculum was removed and replaced with 3 mL BES media and incubated for a further 23 h. The supernatant was harvested and diluted 1 in 10 for subsequent inoculation of the CK cells. The E gene of the resulting passaged isolates was Sanger sequenced at passage 5, 10 and 15 as described in Section 2.11.

### 2.11. Sequencing the E Gene

RNA was extracted from either the cell supernatant or allantoic fluid using the Qiagen RNeasy kit following the manufacturer’s protocol for RNA clean up. RNA was reverse transcribed using random primer, 5′ GTTTCCCAGTCACGATCNNNNNNNNNNNNNNN 3′ and a SuperScript IV reverse transcription kit following the manufacturer’s protocol (Life Technologies). The E gene was amplified using recombinant Taq Polymerase (Life Technologies) using primers 5′-GCTGAAGATTGTTCAGGTGA-3′ and 5′-GCTGAACTGACTGTTCAAAG-3′. The PCR products were Sanger sequenced by Eurofins, and the resulting sequencing data was analysed using the Staden Package 2.0.0b11.

### 2.12. Imaging of Cytopathic Effects (CPE)

CK cells seeded into 6-well plates were inoculated with 1 × 10^5^ PFU of Beau-R, BeauR-T16A or BeauR-A26F isolates or mock infected with BES media. At 24 hpi, the inoculated CK cells were imaged with an AMG EVOS^TM^ XL Core microscope.

### 2.13. Cell Viability Assay

Cells were seeded to confluency in 96-well plates and washed once with PBS prior to inoculation with a two-fold serial dilution, in BES medium, of Beau-R, BeauR-T16A or BeauR-A26F isolates starting at 1 × 10^5^ PFU. Mock infected cells were inoculated with the BES medium. At 24 h intervals, the viability of the cells was assessed using the luciferase assay CellTiter-Glo^®^ kit (Promega), which measures the quantity of ATP present to quantify the number of viable cells following the manufacturer’s instructions, detailed in [41]. The percentage of cell viability determined by the level of luminescence was calculated in comparison to the mock infected cells.

### 2.14. Assessment of Innate Immune Response by Real Time Quantitative PCR (qRT-PCR)

CK cells were seeded to confluency in 6-well plates and were washed once with PBS. Cells were mock infected with BES media or inoculated with 500 µL of Beau-R, BeauR-T16A or BeauR-A26F isolates at a titre of 1 × 10^5^ PFU in BES medium. At 6 and 24 hpi, the cells were harvested, and RNA extracted using the Qiagen RNeasy kit including an on-column DNAse treatment, following the manufacturer’s (Qiagen) instructions. Reverse transcription was carried out using 1.25 µg of total RNA using Superscript IV reverse transcription kit according to the manufacturer’s protocol using random primer (5′ GTTTCCCAGTCACGATCNNNNNNNNNNNNNNN 3′). The resulting cDNA was diluted to ensure 100 µg was added per qRT-PCR reaction. TaqMan qPCR reagents were used to perform qPCR, using either TaqMan ^®^ Fast Advance Master Mix or TaqMan ^®^ Multiplex Mastermix (Life Technologies). Primers were used at 10 μM, and hydrolysis probes were diluted to 5 μM. A GeNorm was carried out and Beta-Actin was selected as an endogenous control; this control has been used in CK cells previously [42]. Sequences for the primers and probes used are listed in Table 1. The qPCR reaction was run on a 7500 Fast Real-Time System following the cycle: 95 °C for 20 s and 40 cycles of 95 °C for 1 s at 60 °C for 20 s.

### 2.15. Statistical Analysis

The statistical analysis was assessed using GraphPad Prism 8.0. The standard deviation and normality were assessed prior to any statistical analysis.

## 3. Results

### 3.1. The Generation of rIBVs Containing Either the T16A or A26F Mutation within the E Protein

The E protein is divided into three domains; a short hydrophilic N terminal domain, a transmembrane domain, which is the focus of this study, and a long hydrophilic C terminal domain [43]. A comparison of the amino acid sequence between a range of IBV strains, representing diverse serotypes as well as genotypes, identified that the sequence of the transmembrane domain of the E is relatively conserved (Figure 1). All IBV strains analysed possessed the residue threonine (T) at amino acid position 16 and alanine (A) at amino acid position 26. The conservation of the amino acid sequence particularly surrounding the T16 and A26 residues highlights that the non-pathogenic laboratory strain, Beau-CK, is a representative IBV strain with regards to the E protein. Therefore, using a vaccinia-virus-based reverse genetics system based on Beau-CK [29], two rIBVs were generated containing individual amino acid changes in the E protein that have been reported to abolish IC activity [16]. The first, denoted by BeauR-T16A, contained a single-point mutation A24246G resulting in the amino acid change T16A. The second rIBV, denoted by BeauR-A26F, contains three-point mutations, GCA to TTT at positions 24276 to 24278, generating the single amino acid change A26F. Both BeauR-T16A and BeauR-A26F were successfully rescued and propagated in primary CK cells. Three isolates of each, from independent rescues, were generated, denoted by BeauR-T16A 1, 2 or 3 and BeauR-A26F 1, 2 or 3. Viral stocks were generated via passaging three times in CK cells for BeauR-A26F and four times in CK cells for BeauR-T16A. The ability to successfully rescue rIBVs with mutations T16A or A26F within the E protein suggests that these residues are not required for viral replication in vitro. This is in line with earlier reports [16,18].

### 3.2. Next-Generation Sequencing Identified an Additional Mutation in One Isolate of BeauR-T16A and One Isolate of BeauR-A26F

Complete genome sequences of the three isolates representing rIBVs BeauR-T16A and BeauR-A26F were assembled using NGS technologies. Consensus genomic sequences generated from stock viruses of all three isolates of both BeauR-T16A and BeauR-A26F were assembled and compared to the Beau-CK reference sequence (GenBank Accession number AJ311317). The stocks of BeauR-T16A and BeauR-A26F were generated at passages 4 and 3 in CK cells, respectively. In all isolates of both BeauR-T16A and BeauR-A26F, the consensus sequences generated confirmed the presence of the mutations T16A and A26F, respectively (Table 2). In BeauR-T16A-2, a synonymous mutation, T13658C, was identified in nsp12. One synonymous mutation in BeauR-A26F-2, T2628C, was identified within nsp3. As the NGS analysis was carried out on viruses that had been passaged, it cannot be determined whether the additional mutations identified arose during the rescue process or the passaging for viral stock generation.

### 3.3. T16 and A26 Residues within the E Protein Are Not Essential for Virus Replication In Vitro

The replication of all isolates of BeauR-T16A and BeauR-A26F were investigated in primary CK cells (Figure 2A,B). All the rIBVs exhibited largely comparable replication to the parental Beau-R from 8–11 hpi and from 24–96 hpi, although some minor differences were observed. The titres of Beau-R were higher than all isolates of BeauR-A26F at 72 hpi (Figure 2B), and, interestingly, BeauR-T16A-3 exhibited lower titres than both the other BeauR-T16A isolates and Beau-R (Figure 2).

The Beaudette strain of IBV has extended cell tropism [44,45]. The replication of the rIBVs was, therefore, also assessed in DF1 cells, an avian cell line derived from chicken embryo fibroblasts [31] and in Vero cells. Vero cells are a mammalian cell line derived from an African green monkey, and, although not directly relatable to natural infection of chickens, they are used for IBV research due to the easier availability of reagents and additionally are of interest as they are licenced for vaccine manufacturing [46,47]. Replication of all three isolates of BeauR-T16A was found to be comparable to Beau-R in both Vero and DF1 cells (Figure 2). Isolates of BeauR-A26F showed comparable replication to Beau-R in DF1 cells (Figure 2C) but not within Vero cells (Figure 2D). In this experiment it must be noted that the replication of the Beau-R control reached almost 10^9^ PFU, which, although unusual, has previously been observed [45]. The reduced replication of BeauR-A26F in Vero cells but not DF1 or CK cells, may suggest that the effect of the A26F mutation is dependent on cell type.

### 3.4. BeauR-T16A and BeauR-A26F Exhibit Reduced Plaque Size in CK Cells

Beaudette rIBVs containing either the T16A or A26F mutations showed reduced plaque size in comparison to WT in Vero cells [18]. Conversely, earlier work by the same group, also in Vero cells, found that the plaque size for the rIBVs containing either the T16A or A26F mutations was comparable to WT [16]. The plaque morphology and size exhibited by all isolates of BeauR-T16A and BeauR-A26F were therefore assessed in chicken cells, specifically primary CK cells and compared to Beau-R (Figure 3). Whilst the plaque morphology appeared comparable, the diameters of the plaques generated by all isolates of BeauR-T16A and BeauR-A26F were smaller than that of Beau-R. This reduction in plaque size may suggest impaired viral spread for viruses with T16A or A26F mutations, as previously hypothesised [18].

### 3.5. BeauR-A26F May Exhibit Impaired Viral Release within Continuous Cell Lines but Not within CK Cells

To et al. indicated that, in Vero cells, the T16A and A26F mutations impede viral release [16]. Research using HeLa cells also showed that the presence of the A26F mutation results in reduced VLP production [12]. To establish whether the T16A and A26F mutations impede viral release, and therefore spread, in biologically relevant cells, the cell lysate and supernatant were harvested from virus-infected CK cells at the peak of viral infection, as determined by the peak in titre as observed in Figure 2. The quantities of infectious progeny in the cell lysate and supernatant were assessed (Figure 4A). An increase in the viral titre of the cell lysate in comparison to the supernatant suggests that the infectious progeny is not effectively being released from the cell, therefore implying a deficiency in the release stage of the replication cycle. Vero cells were also investigated to assess whether any cell-type-dependent differences in results occurred (Figure 4B). It must be noted that the peak titres of IBV replication differ between CK and Vero cells with the peaks observed at 24 and 72 hpi, respectively (Figure 2). Supernatant and cell lysate were therefore harvested at 24 hpi for CK cells and 72 hpi for Vero cells. Cells were inoculated with either Beau-R, BeauR-T16A-1 or BeauR-A26F-3. The latter two isolates were chosen, as no differences were observed between the replications of these isolates and the other isolates of either BeauR-T16A or BeauR-A26F, respectively, as shown in Figure 2. Progeny viruses present within the supernatant and cell lysate were quantified by plaque assay. For CK cells, there was no difference in viral titres between the harvested cell lysates or supernatants, suggesting that neither the T16A nor the A26F mutation were impeding viral release (Figure 4A). For Vero cells, however, higher titres of BeauR-A26F were detected in the cell lysate in comparison to the supernatant (Figure 4B). The reduction observed may suggest that, in this cell type, the A26F mutation is affecting viral release and, therefore, may impede viral spread.

### 3.6. Amino Acid Residues T16 and A26 Are Not Essential for Virus Replication in Ex Vivo TOCs

In vivo, IBV replication primarily occurs in ciliated tracheal epithelial cells [8]. TOCs are an ex vivo culture system consisting of sliced tracheal rings prepared from 19-day-old embryos that offer a more representative model for natural infection than traditional cell cultures. Ciliary activity is reduced in the ciliated tracheal cells of the TOCs following infection with IBV, with the complete cessation termed ciliostasis; this reduction occurs during a natural IBV infection [48]. The reduction in ciliary activity caused by IBV in infected chickens is commonly used in both research and industry to determine the presence of pathogenic IBV [32,48] and can be subsequently used as an indicator of the amount of replicating virus. To investigate the effect of the T16A and A26F mutations on IBV induced reduction in ciliary activity, replicates of 10 ex vivo TOCs were inoculated with either BeauR-T16A, BeauR-A26F, Beau-R or mock infected with media only (Figure 5A). The ciliary activity was observed at 24 h intervals and was comparable between both rIBVs and Beau-R, suggesting comparable replication kinetics. To confirm this, one isolate of each rIBV was selected for a replication assay in which the quantity of infectious progeny generated during the infection of ex vivo TOCs was quantified via titration in CK cells. Although BeauR-A26F appears to have a slightly reduced titre at 96 hpi, statistical significance was not reached (Figure 5B). Replication in ex vivo TOCs is not, therefore, affected by the presence of either the T16A or A26F mutations, suggesting these residues are not required for viral replication in trachea, the site of natural IBV infection.

### 3.7. Both BeauR-T16A and BeauR-A26F Generate Revertant Mutations upon Passage in CK Cells Suggesting a Preference to Retain the T16 and A26 Residues In Vitro

To investigate the genetic stability of both the T16A and A26F mutations, four replicates of Beau-R and the three independent isolates of BeauR-T16A and BeauR-A26F were serially passaged 15 times in CK cells. The E gene of progeny viruses was Sanger sequenced at passages 5, 10 and 15 (Figure 6).

All isolates of BeauR-T16A maintained the T16A mutation at passage 5 and generated no other mutations within the E gene; however, mutations were observed at passage 10 (Figure 6A). A single nucleotide change, G24246A, was observed in all four replicates of BeauR-T16A-1, leading to the reversion of the T16A mutation. In all four replicates of BeauR-T16A-2, the T16A mutation was retained but a C24277T mutation (GCA → GTA) was observed resulting in the amino acid change A26V. One replicate of BeauR-T16A-3 also generated this mutation. The A26V mutation was identified in Vero and DF1 passaged isolates and has been shown to recover IC activity [16]. At passage 15, the BeauR-T16A-1 and BeauR-T16A-2 isolates exhibited the same sequence as at passage 10. Interestingly, the BeauR-T16A-3 isolate showed a different profile in two of the replicates. The replicate which contained the A26V mutation reverted, therefore restoring the original T16A sequence. The replicate which stably maintained the T16A mutation at passage 10 maintained the T16A but generated an A17V mutation, which has not been described. The pattern of mutations suggests that, although IC activity was observed to not be required for in vitro replication, there may be a preference to retain it.

The BeauR-A26F isolates showed greater genetic stability than BeauR-T16A. Isolates BeauR-A26F-2 and BeauR-A26F-3 maintained the A26F mutation over all 15 passages and generated no compensatory mutations within the E gene (Figure 6B). BeauR-A26F-1 generated a point mutation at position T24277G to create a F26C mutation in one replicate at passage 10, which has been shown to restore IC activity [16]. At passage 15, the BeauR-A26F-1 replicate with the F26C mutation remained stable, but, additionally, a different replicate generated a point mutation at position T24277C to generate an amino acid change F26S mutation, which has not been described previously.

### 3.8. High Selection Pressure to Maintain E Protein Activity to Facilitate Replication In Ovo

The IBV strain Beaudette exhibits a non-pathogenic phenotype in birds [49]. This is likely the result of serial passage both in vitro and in ovo; the passaging history of the Beaudette strain is unknown, but reports suggest it has been serially passaged 100–300 times [50]. The Beaudette strain has, however, retained pathogenicity in ovo [18]. Previous research suggested that the IC contributes to the pathogenesis of IBV in embryonated eggs, as the recombinant virus expressing the T16A mutation exhibited a lower 50% embryonic lethal dose (ELD_50_) [18]. To investigate this further, in replicates of three, 10^4^ or 10^5^ PFU of either Beau-R, BeauR-T16A or BeauR-A26F were inoculated into allantoic cavities of 10-day-old SPF embryonated hen’s eggs. The quantity of infectious progeny present, within the allantoic fluid at 24 hpi was determined via titration in CK cells. Additionally, the sequence of the E gene of the isolated passaged viruses was determined via Sanger sequencing.

In contradiction to previous research [18], all isolates of BeauR-T16A exhibited comparable titres to Beau-R (Figure 7A), suggesting comparable replication kinetics. Sequence analysis identified that all replicates of the BeauR-T16A isolates had generated compensatory or revertant mutations, with each isolate generating the same mutation in each replicate. Isolate 1 reverted the T16A mutation, while isolates 2 and 3 generated the compensatory A26V mutation. All isolates of BeauR-A26F displayed statistically lower titres of infectious progeny in comparison to the parental Beau-R (Figure 7A), and, unlike BeauR-T16A, the BeauR-A26F isolates showed stability over the E gene after passage in ovo (Figure 7B). The reduced replication paired with the maintenance of the A26F mutation may therefore suggest that the A26 residue within the E protein is advantageous for in ovo replication. The rapid acquisition of revertant or compensatory mutations observed in the BeauR-T16A isolates, which likely restored replication to a level comparable to Beau-R, may also suggest the T16 residue is advantageous for in ovo replication.

### 3.9. Infection with BeauR-T16A Results in Reduced CPE in Primary CK Cells

To further examine the effect of the T16A and A26F mutations in a relevant cell type, CK cells were inoculated with 10^5^ PFU of either Beau-R, BeauR-T16A or BeauR-A26F, and the CPE was observed. CK cells infected with the IBV show a prominent CPE, including syncytium formation (the fusion of virus infected cells with neighbouring cells resulting in a multinucleated cell) [51], cell rounding, and detaching from the cell culture dish [52]. Syncytia formation is not observed during the infection of cell cultures with all strains of IBV; however, Beau-R does cause extensive syncytia [52]. In CK cells, there was no observed difference in the CPE induced by Beau-R or BeauR-A26F. However, there was a reduction in the level of syncytia formation in BeauR-T16A infected cells in comparison to either Beau-R or BeauR-A26F (Figure 8), suggesting the T16A mutation may have affected the host response to viral infection or viral protein processing, resulting in less fusogenicity.

### 3.10. Neither Infection with BeauR-T16A or BeauR-A26F Impacts CK Cell Viability

The reduced CPE induced by BeauR-T16A was further investigated using a luciferase assay, CellTiter-Glo^®^, which measures ATP levels to quantify the number of metabolically active, and therefore viable, cells [41]. CK cells were inoculated with BeauR-T16A, BeauR-A26F or Beau-R at 1 × 10^5^ PFU and serially diluted twofold. Cell viability was measured at 24 h intervals. At all the time points assessed, the cell viability was observed to be comparable between both rIBVs and the parental Beau-R, indicating infection results in equivalent cell cytotoxicity (Figure 9A).

The same set-up was repeated in DF1 and Vero cells. Interestingly, differences were observed within these continuous cell lines (Figure 9B,C). DF1 cells (Figure 9B) infected with BeauR-T16A exhibited lower cell viability over all titres at 48 and 72 hpi than either BeauR-A26F or Beau-R; however, statistical significance was not reached. At 96 hpi, Beau-R showed lower cell viability when infected with higher titres (ns). Vero cells (Figure 9C) infected with >10^4^ PFU of both BeauR-T16A and BeauR-A26F did not exhibit a reduction in cell viability comparable to Beau-R at 48–96 hpi, with statistical significance reached at 96 hpi. This indicates that the cytotoxic effects of both BeauR-T16A and BeauR-A26F are, therefore, not only dependent on cell type, but may also be dependent on the quantity of virus. This result may also have been influenced by the level of virus production present in Beau-R infected Vero cells in comparison to BeauR-A26F (Figure 2D).

### 3.11. rIBVs Upregulate Innate Immune Factors Comparably to Parental Beau-R

The E protein IC has been shown to be a pathogenicity factor in *Betacoronavirus* SARS-CoV, and mice inoculated with IC inactivated SARS-CoV exhibited decreased levels of Interleukin (IL)-1B and IL-6 [20]. In IBV, IC inactive viruses showed decreased upregulation of IL-6 in Vero cells and 10-day old embryonated eggs, indicating to the authors a role for the IC in the host’s innate immune response to IBV infection [18]. The upregulation of four innate immune factors, Interferon (IFN)-α, IFN-β, IL-6 and IL-1B, within the cell lysate of CK cells infected with either BeauR-T16A, BeauR-A26F or Beau-R (Figure 10) was examined. These factors were selected as they have previously been shown to be upregulated during IBV infection [53,54,55,56]. qRT-PCRs determined comparable levels of IBV-derived RNA between samples, suggesting comparable replication kinetics (Figure 10A). Comparable levels of IFN-α IFN-β, IL-6 and IL-1B upregulation within cells infected with rIBVs containing either a T16A or A26F mutation and Beau-R were also observed (Figure 10B). Whilst it may appear that the rIBVs with a T16A or A26F mutation show slightly higher levels of upregulation of IFN-α and IFN-β than parental Beau-R, statistical significance was not reached.

## 4. Discussion

The ability to successfully rescue rIBVs with either the mutation T16A or A26F within the E protein suggests that these residues are not required for viral replication in vitro, in line with earlier reports [16,18]. It must be noted that work by To et al., also introduced the T16A or A26F mutation into a Beaudette-based backbone [16]. Replication was largely comparable between the two rIBVs and the WT virus in the mammalian cell line, Vero, although the plaque size was reduced [18]. In this study, we generated comparable rIBVs within Beau-R using our independently generated reverse genetics system [29], denoted by BeauR-T16A and BeauR-A26F. The rIBVs in this study were, unlike those generated by To et al. [16], recovered and passaged in chicken cells and the replication kinetics assessed in primary chicken cells and chicken-derived ex vivo cultures. The replication of both BeauR-T16A and BeauR-A26F was comparable to that in the parent virus, Beau-R, in primary CK cells (Figure 2A,B), DF1 cells, a continuous chicken cell line (Figure 2C), and in ex vivo TOCs (Figure 5B), suggesting that these residues are not required for the replication of IBV. This is in line with previous reports not only of IBV [16], but also of other coronaviruses, including SARS-CoV [20].

The replication of BeauR-A26F was reduced in Vero cells in comparison to unmodified Beau-R (Figure 2D and Figure 4C). This is in contradiction with research by To et al., which identified comparable growth kinetics in the rIBVs containing the A26F mutations and WT virus [16]. The differences in findings between the studies may be the consequence of sequence differences between the Beaudette isolates used, in which the rIBVs are based and/or may be the result of difference in batches or lineage of Vero cells. In this study, in Vero cells, a subsequent analysis demonstrated that a higher quantity of infectious progeny was detected in cell lysate rather than supernatant (Figure 4B). Previous research has shown that the hydrophobic domain, in which the T16A and A26F mutations are present, is required for the efficient release of virions [17]. Additionally, the reduced release of infectious virus from Vero cells showed that this reduction in released virus was associated with the IC-inactivating mutations [17]. This, however, was not observed in CK cells, suggesting that, along with the differences in replication kinetics (Figure 4A), the phenotypic effect of either the T16A or A26F mutation may be dependent on cell type. Additionally, in this study, the effect was also not observed with BeauR-T16A, suggesting that the two mutations have differential phenotypic effects and may exert such effects through different mechanistic actions.

The potential for the different mechanistic actions of either the T16A or A26F mutations is further illustrated, as BeauR-A26F exhibited reduced replication in ovo in comparison to both BeauR-T16A and Beau-R (Figure 7A). Sequencing of the E gene of the in ovo passaged viruses, however, identified that all isolates of BeauR-T16A either contained the revertant mutation, A16T, or the compensatory mutation, A26V (Figure 7B). In contrast, no reversion or compensatory mutations were identified in the sequence of all isolates of BeauR-A26F. Rather than differences in mechanistic action, this may simply be the result of the fact that only one nucleotide change is required for the reversion of the T16A mutation (G24246A), but three-point mutations are required to revert the A26F (TTT 24276-8 GCA). Previous research identified that both the T16A and A26F mutations were unstable after five passages in either DF1 or Vero cells, not only highlighting a preference to retain these residues, but also highlighting the mutagenesis ability of the T16A and A26F residues [16]. Conversely, in this study, the passaging of BeauR-A26F isolates 2 and 3 in CK cells identified that the A26F mutation was stably maintained at passage 15, and only two replicates of BeauR-A26F isolate 1 had a mutation at residue 26, T24277G and T24277C, resulting in the amino acid change of F26C and F26S, respectively (Figure 6B). Passage 10 of BeauR-T16A identified either compensatory or revertant mutations in all isolates (Figure 6A). This data (Figure 6) in partnership with the assessment of replication kinetics (Figure 2) and alongside previous research [16] suggest that, while the T16 and A26 residues may not be required for viral replication, there is a selection pressure to retain it. The data also suggests that there is a different selection pressure exerted by the differing cell culture systems, including in ovo. This not only highlights the cell-dependent effect of the T16A and A26F mutations, but may also suggest a cell-dependent role of the E protein itself. Additionally, work on SARS-CoV-2 has also shown cell-type-dependent selection pressure. This work found that a deletion of the cleavage site within the S gene arose upon passage in Vero-E6 cells, which is not dominant in clinical samples [57].

The E protein is thought to play a key role in the assembly [12,14,15] and release [16,17,18] of infectious virions. The E protein IC neutralises and subsequently dissociates the Golgi to ensure the correct processing of viral proteins [17]. The T16 residue has been demonstrated to be required for this dissociation; however, this research was completed in HeLa cells, a human cell line [12]. The cleavage and processing of the S2 subunit take place within the Golgi apparatus, and it been shown that the presence of the T16A mutation subsequently results in the incorrect processing of the S2 subunit [28]. Syncytia formation (which allows for humoral immune evasion, as it enables viruses to spread between cells without having to enter the extracellular environment, therefore shielding progeny virions from neutralising antibodies [58] is facilitated by the S2 subunit of the S protein of IBV [45]. Lower levels of syncytium were observed during BeauR-T16A and BeauR-A26F infection of CK cells (Figure 9), which may suggest that the mutations have resulted in the incorrect processing of the S2 subunit. Furthermore, both BeauR-T16A and BeauR-A26F exhibit smaller plaque phenotypes than Beau-R (Figure 3), as demonstrated in Vero cells [18]. This may be the result of reduced syncytium formation negatively impacting the viral spread between cells.

A classical characteristic of viroporin activity is the modulation of the host response to infection. It has been shown that IC inactive mutants of both SARS-CoV [20] and IBV [18] exhibit differential upregulation of the expression of innate immune genes in comparison to WT. In this study, the upregulation of four innate immune factors was shown to be comparable between both BeauR-T16A and BeauR-A26F, as well as the parental Beau-R (Figure 10) in CK cells. This is contrary to research in ovo and in Vero cells, which showed a reduction in innate immune upregulation when IC inactivating mutations were present [18]. The differing results obtained from this work are likely due to the use of primary CK cells in comparison to other culture systems. It must be noted that Vero cells have a restricted innate immune response [59], and chick embryos at 10 days have a limited innate immune response, which is not fully established until day 18 of embryonic development [60,61]. The differing results observed, therefore, may suggest further that both the T16A and A26F mutations have a cell-dependent effect. Interestingly in this study, different cytotoxicity effects were observed in Vero cells infected with either BeauR-T16A or BeauR-A26F, with a significantly lower level of cytotoxicity observed compared to Beau-R. This effect was not observed with either CK cells or DF1 cells, further indicating the cell-dependent effect of the T16A or A26F mutations and, therefore, possibly the E protein itself. This is not the first study to indicate cell-type-dependent results. Cell-type-dependent virus–host interactions have been identified in the study of autophagy, with IBV inducing autophagy in Vero cells but not within avian cell lines, DF1 and CK cells [62].

## 5. Conclusions

In conclusion, we have generated and characterised the T16A and A26F mutations within the hydrophobic domain of the E gene in the rIBV Beau-R. The rIBVs BeauR-T16A and BeauR-A26F are replication-competent, demonstrating that these residues are not essential for viral replication. The genetic stability of the T16A and A26F mutations, the replication of the rIBVs and the cytopathogenicity induced were assessed in a variety of systems, ranging from avian primary cell lines, continuous cell lines of both avian and mammalian origin, and in ovo and avian ex vivo TOCs. Differing results were observed through the assessment of the same parameter within different systems, highlighting the importance of cell-type or cell-system selection during IBV E protein research. Further studies in biologically relevant systems are therefore required to further elucidate the role of the IBV E protein during infection.

## Figures and Tables

**Figure 1 viruses-14-01784-f001:**
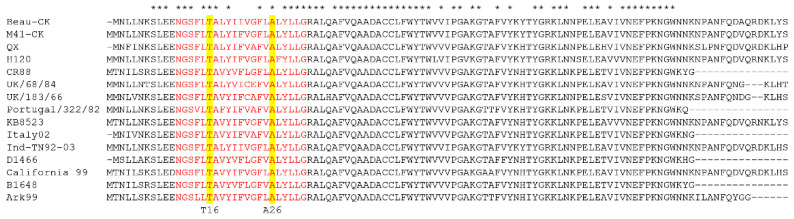
Alignment showing the amino acid sequence similarity between IBV strains over the transmembrane domain of the E protein, highlighted in red. Amino acid sequence alignment over the E protein of different strains of IBV, showing the T16 and A26 residue present in each strain highlighted in yellow. The accession number of strains included were Beau-CK (CAC39117.1), M41-CK (QCE31536.1), QX (ARI46255.1), H120 (UQM93960.1), CR88 (QKV27915.1), UK/68/84 (P30247.1), UK/183/66 (P30248.1), Portugal/322/82 (P30246.1), KB8523 (P19744.1), Italy02 (QKV27954.1), Ind/TN92/03 (YP_009825001), D1466 (QKV27928.1), California/99 (AAS00083), B1648 (ALH21114) and Ark99 (AAX39774). The alignment was assembled on Mega11 using the MUSCLE alignment tool, * represents conserved amino acid.

**Figure 2 viruses-14-01784-f002:**
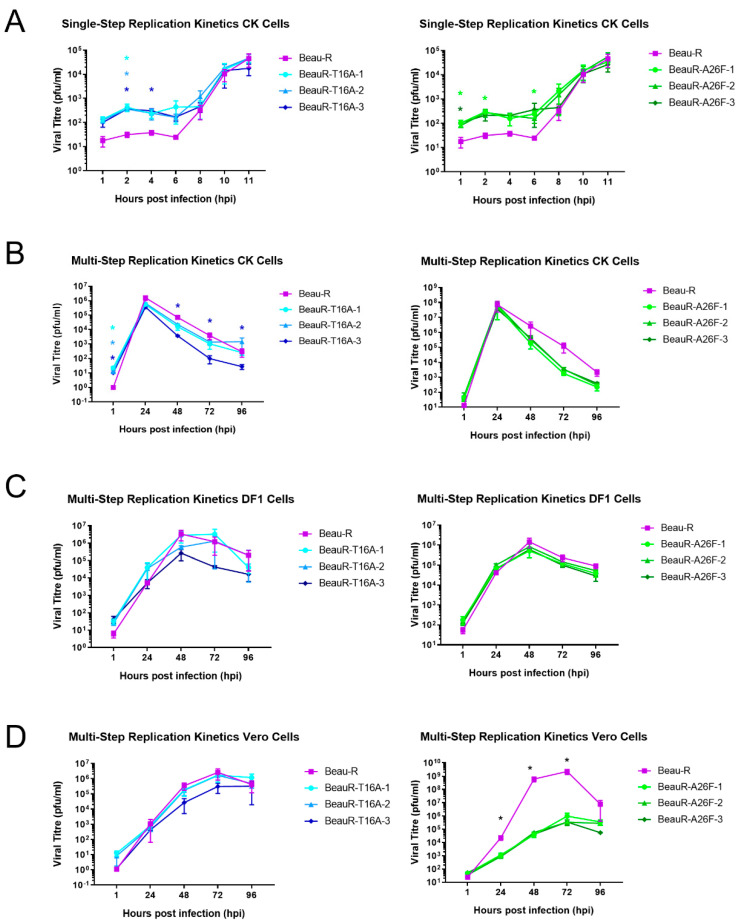
Replication kinetics of rIBVs differ between cell type. (**A**) Single-step replication kinetics of rIBVs in CK cells. (**B**) Multi-step replication of rIBVs in CK cells. (**C**) Multi-step replication kinetics of rIBVs in DF1 cells. (**D**) Multi-step replication kinetics of rIBVs in Vero cells. CK, Vero or DF1 cells were infected at a titre of 1 × 10⁴ PFU (MOI~0.008) for isolates of BeauR-T16A and 1 × 10⁵ (MOI~0.08) for isolates of BeauR-A26F. For multi-step replication kinetics, the supernatant was harvested at 1, 24, 48, 72 and 96 hpi. For single-step replication kinetics, the supernatant was harvested at 1, 2, 4, 6, 8, 10 and 11 hpi. Supernatant was titrated in triplicate on CK cells to determine the quantity of progeny virus. Error bars represent ± standard error of the mean (SEM) of three independent experiments. Statistical analysis was carried out using a two-way analysis of variance (ANOVA), with significance taken at *p*-value of <0.05, significance is represented with a * in relation to Beau-R and for (**A**,**B**) significance is represented with a * with different colours corresponding to the different isolates.

**Figure 3 viruses-14-01784-f003:**
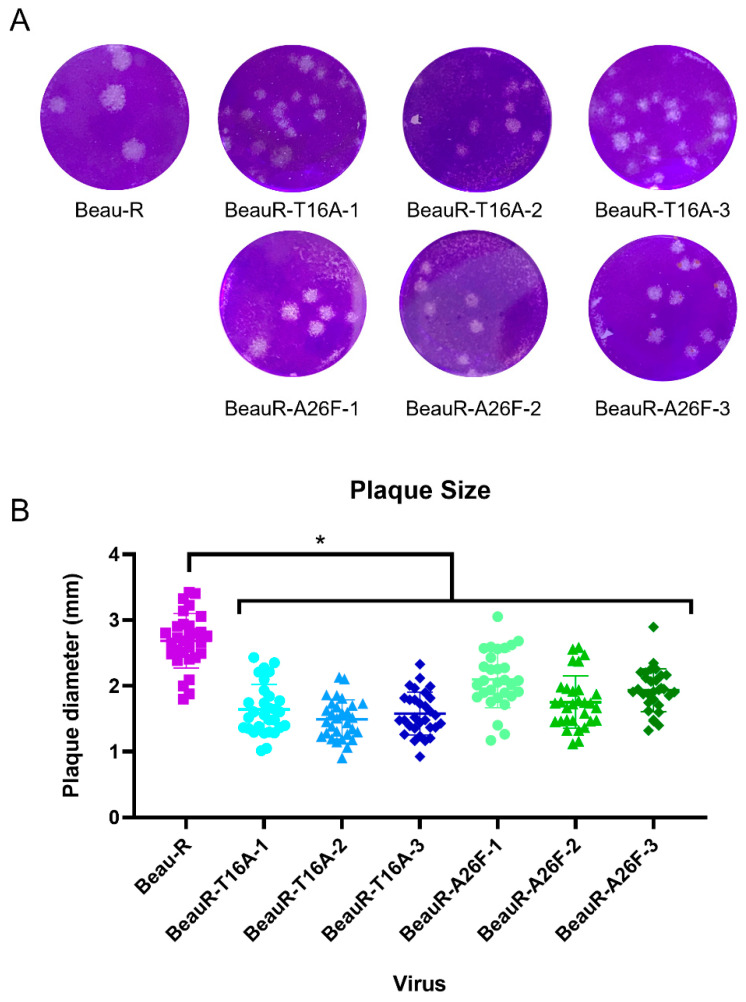
BeauR-T16A and BeauR-A26F exhibit smaller plaque size than parental Beau-R. (**A**) Representative images of plaques formed by each of the rIBVs. (**B**) Plaque diameter measured using ImageJ software with 30 plaques counted per virus, 10 plaques were counted per biological repeat. Error bars represent the standard deviation (SD) of the 30 different plaque sizes. Statistical analysis was carried out using a one-way ANOVA with significance taken as *p*-value < 0.05, represented as *. Significance shown is in relation to Beau-R.

**Figure 4 viruses-14-01784-f004:**
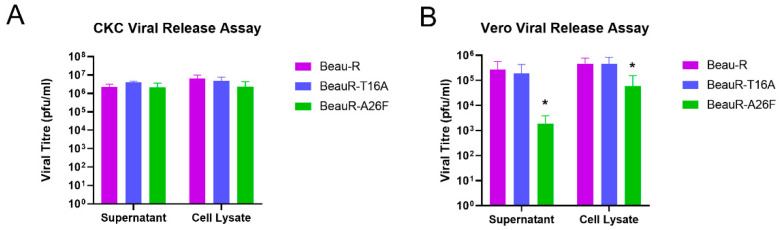
BeauR-A26F exhibits reduced cell release during infection of Vero cells. Confluent (**A**) CK cells and (**B**) Vero cells seeded in 6 well plates were infected at 1 × 10⁴ PFU (MOI~0.008) of either Beau-R, BeauR-T16A-1 or BeauR-A26F-3. Supernatant and cell lysates were harvested at either (**A**) 24 hpi or (**B**) 72 hpi and the quantity of infectious progeny determined via plaque assay in CK cells. Error bars represent ± SEM of three independent experiments. Statistical analysis was carried out using a one-way ANOVA, significance was taken as *p*-value < 0.05, represented as *. Significance shown in relation to Beau-R.

**Figure 5 viruses-14-01784-f005:**
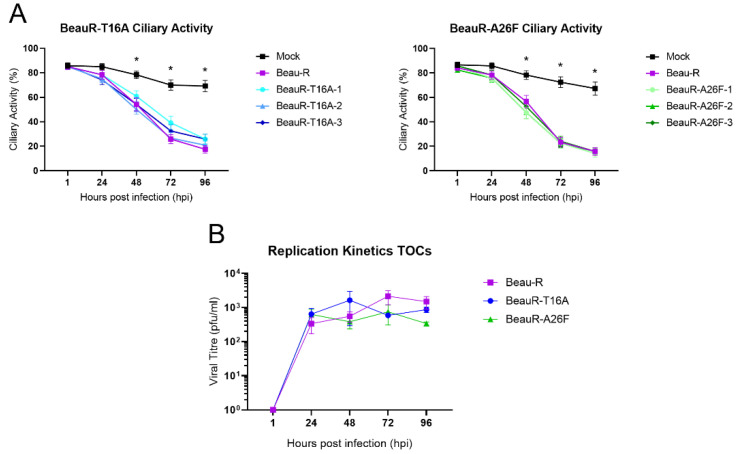
Comparable replication of rIBVs and parental Beau-R in TOCs. (**A**) Ten TOCs were infected with each virus at a titre of 1 × 10⁴ PFU. The ciliary activity was assessed every 24 h from 1 hpi. Each TOC ring was given a score of either 0, 25, 50, 75 or 100% to reflect the proportion of cilia beating at each timepoint. Error bars represent ± SD of three independent experiments. Statistical analysis was carried out using a Friedman test, *p*-value of <0.05 is shown in comparison to mock and indicated by * to represent all viruses. No differences were observed between the rIBVs. (**B**) Six TOCs were infected with viruses at a titre of 1 × 10⁴ PFU. Isolates BeauR-T16A-1 and BeauR-A26F-3 were used for this experiment. Supernatant was harvested over a 96 h time-course every 24 hpi. Virus titre within the supernatant was quantified by titration on CK cells. Error bars represent the ±SEM of three independent experiments. Statistical analysis was carried out using a two-way ANOVA with significance taken as *p*-value of <0.05, no significance was found.

**Figure 6 viruses-14-01784-f006:**
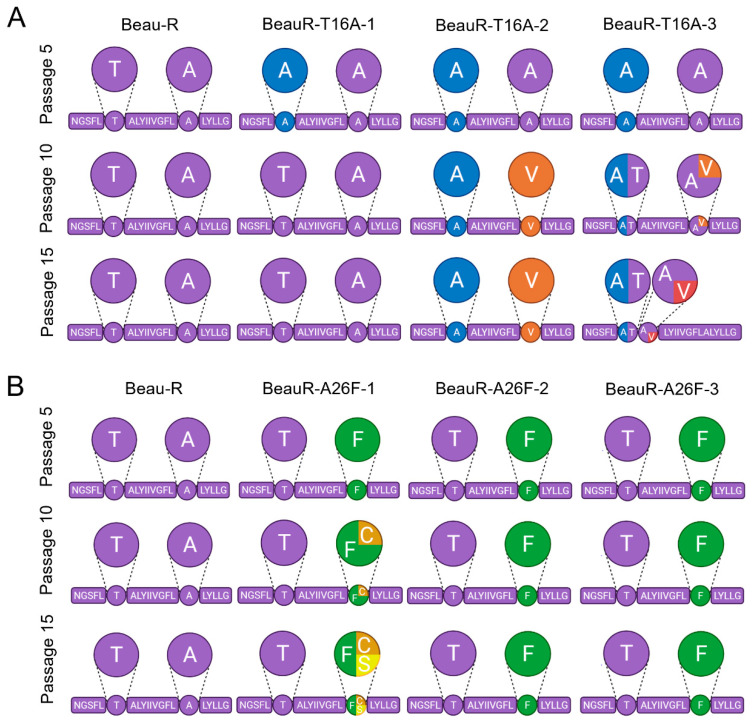
Selective pressure exists to maintain T16 and A26 residues upon passage in CK cells. The transmembrane domain of the E gene is shown, as that is where all mutations were found. The circles highlight the T16 and A26 residues, except in the case of BeauR-T16A-3 at passage 15 in which A17 is highlighted. Larger circles represent a pie chart of the four replicates of each isolate at each passage showing the sequences present. Every 5 passages, the supernatant was harvested, and Sanger sequenced over the E gene to detect any mutations generated. Mock wells did not contain any virus at each round of screening through RT-PCR. (**A**) Four replicates of each isolate of BeauR-T16A virus were passaged alongside Beau-R and media mock wells for 15 passages, diluted 1 in 10 at each round. The Beau-R sequence is shown in purple, T16A mutation is shown in blue, A26V is shown in orange, and A17V is shown in red. (**B**) Four replicates of each isolate of BeauR-A26F were passaged 15 times alongside Beau-R and mock infected wells. Beau-R sequence is shown in purple, the A26F mutation is shown in green, F26C is shown in orange, and F26S is shown in yellow.

**Figure 7 viruses-14-01784-f007:**
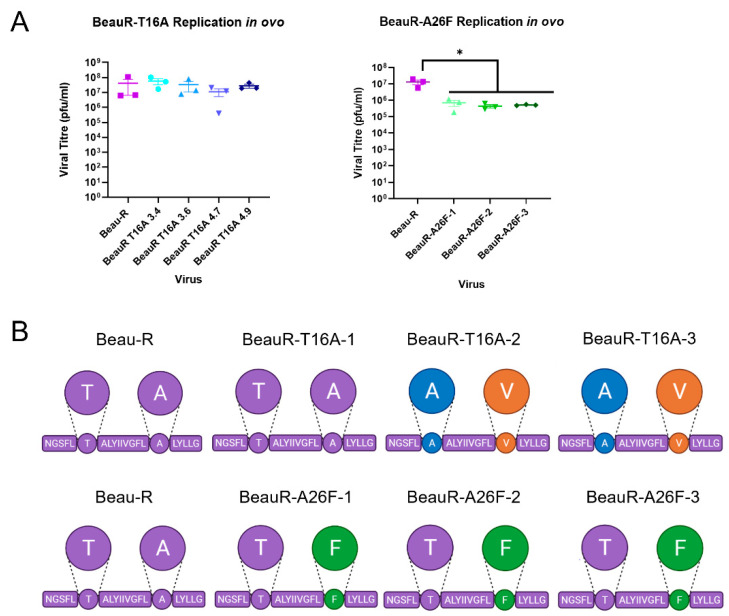
Reduced replication or genetic stability of rIBVs in ovo. Eggs were infected at a titre of 1 × 10⁵ PFU (MOI~0.08) for Beau-R and all isolates of BeauR-T16A and BeauR-A26F. (**A**) The allantoic fluid from the eggs was harvested at 24 hpi and titrated on CK cells to determine the quantity of virus present. Error bars represent ± SEM of three independent experiments. Statistical analysis was carried out using a one-way ANOVA with significance taken as *p*-value < 0.05 in comparison to Beau-R and is indicated by *. (**B**) The schematic represents the transmembrane domain of the E protein with the T16 and A26 residues highlighted with circles. Three replicates of each virus were harvested, and Sanger sequenced over the E gene. Beau-R sequence is shown in purple, T16A is shown in green, A26V is shown in orange, and A26F is shown in green.

**Figure 8 viruses-14-01784-f008:**
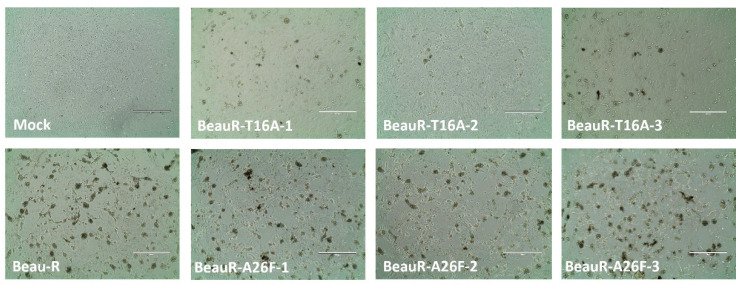
CPE caused by rIBVs. Representative CPE images were taken at 24 hpi on a light microscope for each rIBV. The CK cells were infected with 500 μL of diluted virus at a titre of 1 × 10^5^ PFU (MOI~0.08). Scale bars represent 400 μm.

**Figure 9 viruses-14-01784-f009:**
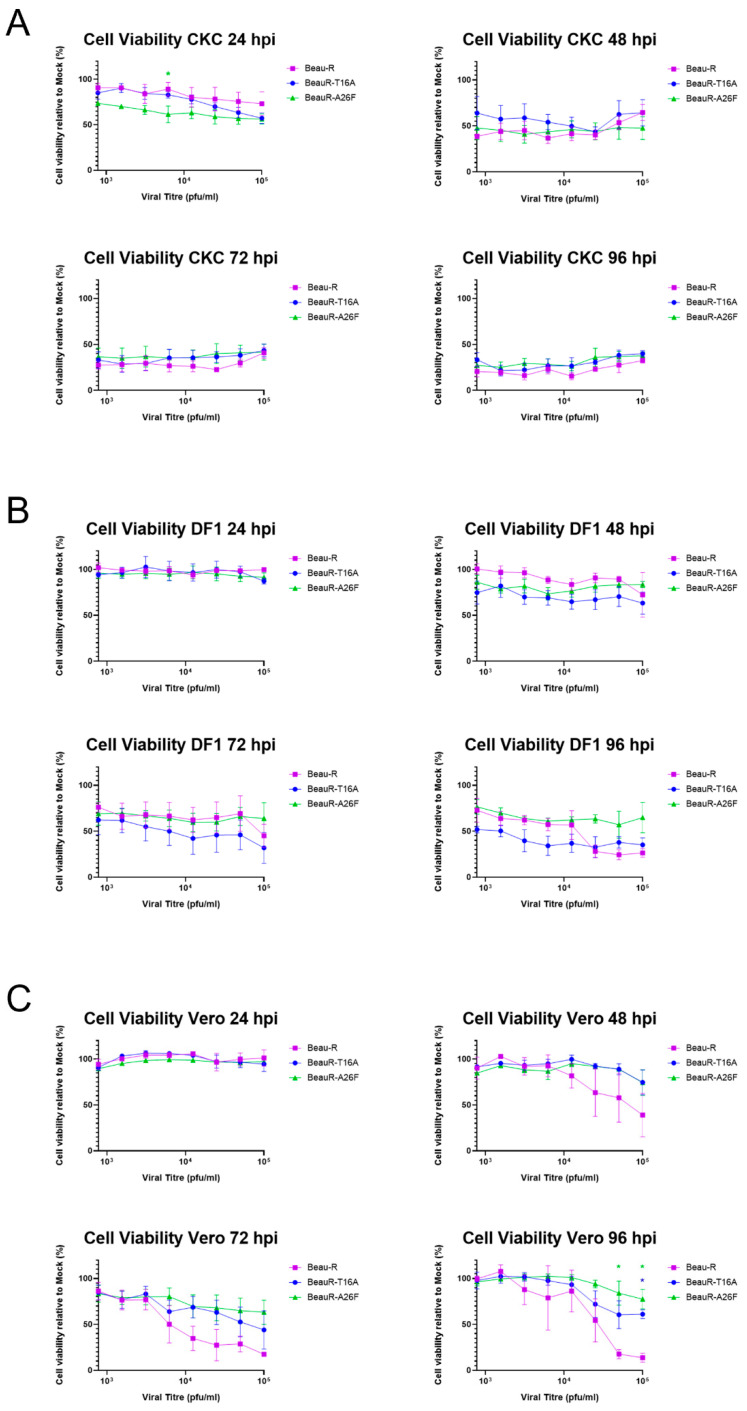
Cell viability reduction differs between cell type. CellTiter-Glo^®^ reagent was used to quantify cell viability within CK (**A**), DF1 (**B**) and Vero (**C**) cells over a 96 h time-course of infection at a range of viral titres. Cell viability was assessed every 24 h. Beau-R, BeauR-T16A-1 and BeauR-A26F-3 were used in this experiment, as well as a cell and media control to account for excess signal. The data is represented as percentage viability in relation to mock-infected cells. Error bars represent ± SEM of three independent experiments. Statistical analysis was carried out using a two-way ANOVA with significance taken as *p*-value < 0.05, significance is shown with an *. Green * and blue * represent A26F and T16A in comparison to Beau-R, respectively.

**Figure 10 viruses-14-01784-f010:**
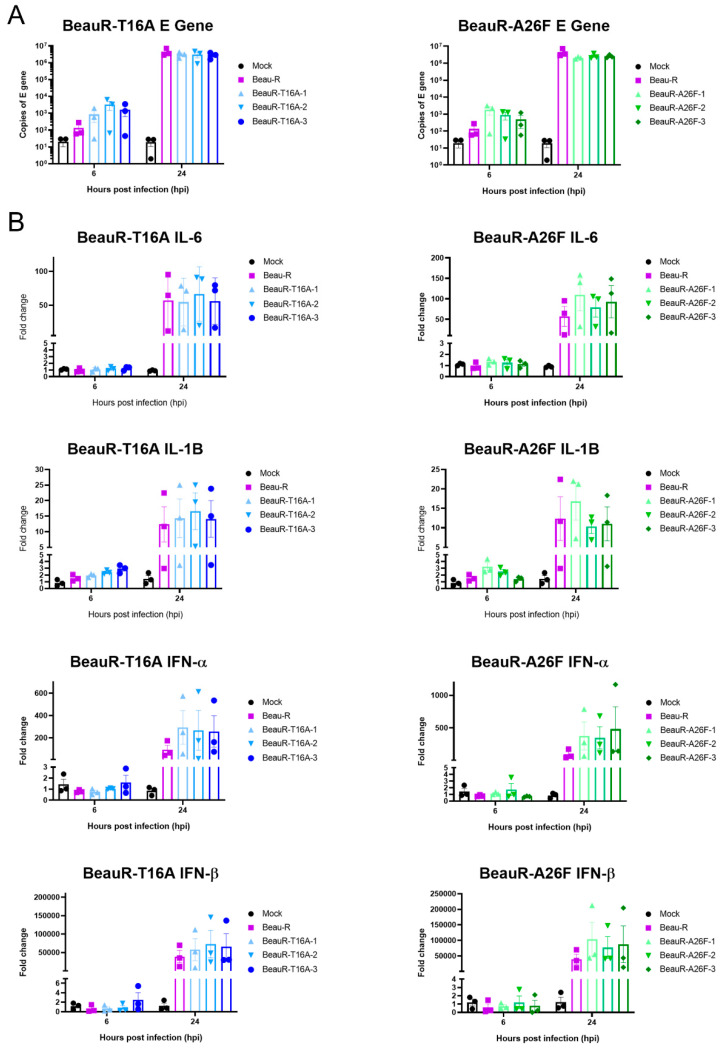
Upregulation of IL-6, IL-1B, IFN-α and IFN-β expression is comparable between parental Beau-R and rIBVs. CK cells were infected with 1 × 10^5^ PFU (MOI~0.08) virus. At 6 and 24 hpi, cells were harvested, and RNA extracted. Error bars represent ± SEM of three independent experiments. (**A**) qPCR representing E transcript present within each sample indicating comparable levels of viral load. (**B**) qPCR investigating upregulation of a range of innate immune factors. Statistical analysis was carried out using a two-way ANOVA and statistical significance was taken as *p*-value < 0.05, no significance was identified.

**Table 1 viruses-14-01784-t001:** Sequences of primers and probes used within qPCR.

Gene		Sequence (5′ to 3′)
E gene	Forward	GGTAGAGCACTTCAAGCATTT
	Reverse	CCGGATTGTTAAGTTTTCTACC
	Probe	CCAGGAGCTAAGGGTACAGCCT
β-Actin	Forward	GCATACAGATCCTTACGGATATCCA
	Reverse	CAGGTCATCACCATTGGCAAT
	Probe	CACAGGACTCCATACCCAAGAAAGATGGC
IL-6	Forward	GCTCGCCGGCTTCGA
	Reverse	GGTAGGTCTGAAAGGCGAACAG
	Probe	AGGAGAAATGCCTGACGAAGCTCTCCA
IL-1B	Forward	GCTCTACATGTCGTGTGTGATGAG
	Reverse	TGTCGATGTCCCGCATGA
	Probe	CCACACTGCAGCTGGAGGAAGCC
IFN-A	Forward	GACAGCCAACGCCAAAGC
	Reverse	GTCGCTGCTGTCCAAGCATT
	Probe	CTCAACCGGATCCACCGCTACACC
IFN-B	Forward	CCTCCAACACCTCTTCAACATG
	Reverse	TGGCGTGTGCGGTCAAT
	Probe	TTAGCAGCCCACACACTCCAAAACACTG

**Table 2 viruses-14-01784-t002:** NGS sequence data showed either T16A or A26F mutations were present at consensus level within rIBVs.

rIBV	Isolate	Position	Ref. nt	Alt. nt	Depth	Freq. (%)	Aa	Gene
BeauR-T16A	1	24246	A	G	22067	87.3023	T16A	E
	2	**13658**	**T**	**C**	**528**	**92.803**	**Y451Y**	**Nsp12**
		24246	A	G	16849	93.0382	T16A	E
	3	24246	A	G	18270	94.4773	T16A	E
BeauR-A26F	1	24276	G	T	8184	99.6823	A26F	E
		24277	C	T	8052	99.6398	A26F	E
		24278	A	T	8071	99.5787	A26F	E
	2	**2628**	**T**	**C**	**464**	**54.9569**	**C29C**	**Nsp3**
		24276	G	T	20386	99.6566	A26F	E
		24277	C	T	20457	99.6432	A26F	E
		24278	A	T	20603	99.6408	A26F	E
	3	24276	G	T	18670	99.6358	A26F	E
		24277	C	T	18857	99.6288	A26F	E
		24278	A	T	18639	99.6083	A26F	E

Notes: BeauR-T16A and BeauR-A26F stocks used to generate this sequence data were at passages 4 and 3 in CK cells, respectively. Nucleotide positions listed are in relation to the Beau-CK genome (GenBank accession number AJ311317). The “Ref. nt” column shows the nucleotide present at that position within Beau-R, and the “Alt. nt” column represents the different nucleotide present within the isolate. The depth shows the number of reads over the position of the mutation, and the allele frequency is the percentage of reads which contained the altered nucleotide. Consensus-level mutations listed were determined by a percentage within the population (% population) of above 50%. Mutations present in other regions of the genome are shown in bold.

## Data Availability

The sequence of Beau-CK has been deposited in GenBank under the accession number AJ311317.

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
