# Peer review of "The Genetic Stability, Replication Kinetics and Cytopathogenicity of Recombinant Avian Coronaviruses with a T16A or an A26F Mutation within the E Protein Is Cell-Type Dependent"

_viruses, 2022, doi:10.3390/v14081784_

Round 1

Reviewer 1 Report

Dear authors,

I have read your manuscript entitled “The Genetic Stability, Replication Kinetics and Cytopatho-genicity of Recombinant Avian Coronaviruses with a T16A or an A26F mutation within the E Protein is Cell Type Dependent.”. The authors demonstrated the significance of the T16A or A26F mutation in E protein in the replication and spread of IBV virus through a large number of experiments. The experimental design is scientific and reasonable, and the results of the article are rich and detailed. However, there are still some problems in the manuscript that need to be fixed. In general, the manuscript should be carefully revised to be considered for further publication.

The following are my comments to your manuscript:

Major comments:

1.      Table 2: How many times the mutant virus has been passaged after rescue? How to prove that the extra point mutations in nsp3 and nsp12 emerged in the rescue process rather than in passage?

2.      In the manuscript, the author detected the viral titre of the supernatant and cell lysate of CK cells and Vero cells respectively. They found that the BeauR-A26F presented higher titres in the cell lysate than the supernatant. They concluded that A26F mutation could affecting viral release and therefore may impede viral spread. However, in the figure 2D, decreased viral titre was observed in the A26F mutant infected Vero cells, which make me concerned that the decrease of virus titre in the supernatant may be the consequence of damaged replication of the mutant virus. From the current experimental results, it is impossible to determine which stage of virus replication the mutation specifically affects, and more experimental data should be provided to support, for example, the virus titer data in the supernatant at different time points can be added in the manuscript.

Minor comments:

1.      Figure 2: The description of the significant difference in the figure legend is a little redundant, so it is recommended to keep only the last sentence.

2.      Figure 2A: Why doesn't figure 2A show 0h data like figure 2B, C and D?

3.     Figure 5: The figure legend showed that the ciliary activity was assessed every 24 hours from 48 hpi, however, 24hpi data has been shown in figure 5A and B. It is recommended that changes be made to maintain uniformity.

4.     Figure 7 was missing in the manucript.

5.     Line 476: The author declared that the T16A and A26F residues within the E protein are advantageous for in ovo replication. I don't quite understand the reason for this conclusion.

Reviewer 2 Report

The manuscript presents the effect of T16A and A26F in the replication of viruses in various culture systems. The role of mutations in ion channel protein for coronavirus replication is important for understanding the mechanism of virus replication. The study is well designed and generally well-written, but a few points need to be addressed before the publication of the manuscript. And the purpose of this study needs to be more clearly described. I have some questions and suggestions below.

1. Abstract, line 24: The study was conducted using IBV, gammacoronavirus. But at the end of the abstract, the conclusion is expanded to ‘coronavirus’. In addition, the BeauR strain is a unique strain that shows different cell tropism compared to other IBVs. Thus, the conclusion should not expand to other coronaviruses.

2. Introduction, lines 81-99: This is a conclusion. It needs to be moved to the discussion or conclusion section. The purpose of this study needs to be described at the end of the introduction.

3. Materials and Methods, line 111: Is the Beau-R strain name? or molecular clone name?

4. Results, lines 292-299: The sentences are hard to understand. Did you use different backbones for generating each rIBVs? If so, it needs to be described in the Methods section. If not, the sentences look not related to this study. The additional mutation can be generated during culturing of viruses.

5. Figure 2, figure 5: If the replication kinetics of BeauR-T16A and BeauR-A26F are shown in a panel, they can be compared to each other, and the size of the figure can be reduced.

6. Figure 2: What are the Single-step replication and Multi-step replication? Do you mean short-term and long-term incubation? Please use clear words or describe it in the Methods section.

7. Results, line 360: peak timing?

8. Figure 7 is missing

9. Results, section 3.8: For the revertant mutation analysis in cells, the author conducted 5, 10 and 15 passages. But the serial passage is not conducted for the revertant mutation analysis in ovo. In addition, 24h incubation is usually not sufficient for virus incubation and not enough for detecting revertant mutations.

10. Conclusion: As I mentioned in the Abstract, the conclusion should not expand to other coronaviruses.
